# Modeling and Forecasting the Volatility of NIFTY 50 Using GARCH and RNN Models

**Vanshu Mahajan [1], Sunil Thakan [1] and Aashish Malik [2,\*]**

1. Department of Chemical Engineering and Biochemical Engineering, Rajiv Gandhi Institute of Petroleum Technology, Jais 229304, India; eche17057@rgipt.ac.in (V.M.); eche17055@rgipt.ac.in (S.T.)
2. Department of Petroleum Engineering and Geoengineering, Rajiv Gandhi Institute of Petroleum Technology, Jais 229304, India
* Correspondence: epe17001@rgipt.ac.in

**Abstract:** The stock market is constantly shifting and full of unknowns. In India in 2000, technological advancements led to significant growth in the Indian stock market, introducing online share trading via the internet and computers. Hence, it has become essential to manage risk in the Indian stock market, and volatility plays a critical part in assessing the risks of different stock market elements such as portfolio risk management, derivative pricing, and hedging techniques. As a result, several scholars have lately been interested in forecasting stock market volatility. This study analyzed India VIX (NIFTY 50 volatility index) to identify the behavior of the Indian stock market in terms of volatility and then evaluated the forecasting ability of GARCH- and RNN-based LSTM models using India VIX out of sample data. The results indicated that the NIFTY 50 index's volatility is asymmetric, and leverage effects are evident in the results of the EGARCH (1, 1) model. Asymmetric GARCH models such as EGARCH (1, 1) and TARCH (1, 1) showed slightly better forecasting accuracy than symmetric GARCH models like GARCH (1, 1). The results also showed that overall GARCH models are slightly better than RNN-based LSTM models in forecasting the volatility of the NIFTY 50 index. Both types of models (GARCH models and RNN based LSTM models) fared equally well in predicting the direction of the NIFTY 50 index volatility. In contrast, GARCH models outperformed the LSTM model in predicting the value of volatility.

**Keywords:** forecasting; Indian stock market; India VIX; NIFTY 50; leverage effects; GARCH models; LSTM model

## 1. Introduction

The equity market of a country is an important part of its economy. It is one of the most influential investment platforms for companies and investors. The stock market is driven by the emotions of those who participate in it, as their desire to purchase a specific stock creates demand, and their willingness to sell a stock creates supply. These supply and demand dynamics continue to occur in the stock market. Therefore, people are always looking for ways to reduce risk since the riskiness in the stock market is influenced by volatility; consequently, it becomes critical to examine the market's volatility before making any investment. Volatility is a measure of unpredictability that is the consequence of investment choices, risk management, and even a country's monetary policy. The expected market risk premium is positively associated with predictable volatility of stock market returns, according to French et al. (1987). Theoretically, stock return volatility may be traced back to Black's (1976) book, which says that leverage effects are the primary source of stock return volatility. In other words, if a company's equity falls while all other factors remain constant, the debt/equity ratio rises, ultimately resulting in an increment in the risk in that equity. Accurately describing how stock prices fluctuate and calculating the stock market's future rate of return has become a hot topic in academics and the financial world. As a result, it has become critical to forecast the volatility to minimize the risk.

The homoscedasticity assumption appears to be violated for time series with features such as a "sharp peak," "fat tail," and volatility clustering. As a result, traditional econometrics models tend to show significant errors while forecasting the financial time series data, such as in Liu and Morley (2009). To overcome this problem, economic scholars have conducted extensive studies to improve the accuracy of traditional econometric models. Professor Engle devised the auto-regressive conditional heteroskedasticity (ARCH) model to address the issue (1982). The ARCH model defies conventional wisdom by rejecting the linear premise of the risk–return relationship. Instead, this approach employs changing variance to build a function linked to past volatilities, which better describes financial data with characteristics such as a "sharp peak" and a "fat tail." With time, researchers discovered that ARCH required a large order of q to accurately predict conditional heteroscedasticity. So, Bollerslev (1986) presented the generalized auto-regressive conditional heteroskedasticity (GARCH), in which the lag phase was incorporated into conditional variance by using the ARCH model. The GARCH model performs efficiently in capturing the regular fluctuations in the volatility of financial data. As a consequence of its effectiveness in capturing frequent changes in financial data volatility, the GARCH model's use in the financial sector has increased in recent years. Later on, Nelson (1990b) established the necessary and sufficient conditions on the GARCH (1, 1) model and he developed a class of diffusion approximations based on Exponential ARCH model in Nelson (1990a). It is vital to evaluate whether the data are statistically appropriate before feeding it into models. Using data from the S&P 500 daily index, Lin and Yang (2000) established a class of statistical tests for financial models. The residual autocorrelation test can be effective in model diagnostic testing for non-linear time series with conditional heteroskedasticity as shown in Li and Mak (1994).

Researchers have created novel recurrent neural network (RNN)-based approaches such as long short-term memory by Hochreiter and Schmidhuber (1997), which show promise in forecasting financial data, due to recent advances in artificial intelligence. Recurrent neural network (RNN) is a type of machine learning model that deals with sequential data inputs and outputs. RNN captures the temporal link across input/output sequences by delivering feedback to feedforward neural networks. RNN extends the neural network (NN) model by adding recurrent layers. Three sets of layers make up a generic RNN model (input, recurrent, and output). Initially, the input data are transformed into a vector that transmits the input's characteristics via input layers. The recurrent layers, which offer feedback, are then applied. The model then concludes in the same way as other NN models, with fully connected (FC) layers at the end, referred to as RNN model output layers. There are primarily two types of RNN models, i.e., LSTM-based RNN models and GRU-based RNN models (Cho et al. 2014), and these models are distinguished by the recurrent layers utilized in them. The RNN and GARCH models are employed in this study to predict the volatility of the National Stock Exchange Fifty (NIFTY 50) index.

This study utilizes the family of GARCH models to forecast the volatility of the NIFTY 50 index. Through an empirical analysis of the econometric models, the symmetric GARCH model (GARCH (1, 1)) and the asymmetric GARCH model (EGARCH, TARCH) are used in this study to discover the peculiarities in the volatility of the NIFTY 50 index. This paper focuses on the volatility of the whole National Stock Exchange (NSE) and the NIFTY 50 index (combination of top 50 equities of NSE) roughly mirrors the NSE's performance. As a result, it was decided to investigate the NIFTY 50 index's volatility. This research aims to determine if these new and advanced econometric models can capture the NIFTY 50's volatility spikes. The goal of this study is to assess the current health of the Indian stock market, and studying the NIFTY 50 index is an effective approach to do so because the NSE handles the majority of India's day trading. This research also aims to assess the RNN-based LSTM algorithm's capability in capturing the volatility of the NIFTY 50 index and compare its performance with that of GARCH models on out-of-sample data. With the aid of the GARCH models, this study also seeks to uncover asymmetric traits present in index volatility. This study uses only the Indian stock market because, according to

World Bank data, India is ranked 6th among all countries in terms of GDP, with forecasts indicating that it will be ranked 3rd, just behind China and the United States by the year 2050. India is the world's second most populous country, however, there has been little study in utilizing new techniques for predicting the volatility of the Indian stock market, although researchers have conducted substantial research for other major countries such as the United States, China, and Japan.

## 2. Literature Review

The volatility of price changes in financial markets is one of the critical concerns in current economic research. Volatility has been a mandatory risk-management activity for many financial organizations since the first Basle Accord was established in 1996 (Poon and Granger 2003). In the 1960s, academics began to look into the stock market's volatility. Stock market volatility has a clustering characteristic according to Mandelbrot (1963), with significant movements followed by more large swings. Bali et al. (2009) found that there is an intertemporal relationship between downside risk and expected returns. Then, researchers started to develop methods to capture these fluctuations in stock prices. The ARCH method of Engle (1982) offers a distinct benefit in terms of predicting the variability of financial time series. It has been widely employed in analyzing financial markets, including equity markets, foreign exchange, crypto markets, and forex markets. The time series data (stock market prices in this study) are sensitive to different contingent elements from a statistical standpoint; therefore, it appears unpredictable. In the GARCH model proposed in 1986, the factors that influence conditional variance were increased to two aspects: mean square error and conditional variance from prior periods (Bollerslev 1986).

Using the two models mentioned above, the researchers identified asymmetric events in the volatility of financial time series. The variations due to negative news are always considerably higher than those caused by good news; the studies of Black (1976); French et al. (1987); Nelson (1990b); and Zakoian (1994) all support this phenomenon. Fong and Koh (2002) utilized the Markov switching exponential generalized auto-regressive conditional heteroskedasticity (EGARCH) method to identify a substantial asymmetry influence in the Heng Seng index (HSI) during periods of extreme volatility. Engle and Ng (1993) applied GARCH models (symmetric and asymmetric) to analyze the Japanese equity market and proposed the news effect curve. Furthermore, the empirical study revealed that there was volatility asymmetry in the Japanese stock market.

Khan et al. (2019) used GARCH models to conduct an empirical analysis of stock market data from 11 religion dominant countries (RDC) and found that volatility clustering and asymmetric behavior are present in RDC stock markets. The study discovered that the Glosten–Jagannathan–Runkle generalized auto-regressive conditional heteroskedasticity (GJR-GARCH) and EGARCH performed better than GARCH (p, q) in predicting 11 RDCs equity markets. Furthermore, GJR-GARCH and EGARCH are utilized to explain asymmetric behavior which resulted in volatility clustering observed in the studied stock markets. From 21 May 1992 to 2 February 1996, Song et al. (1998) examined the Shanghai Stock Exchange (SSE) index and the Shenzhen Composite (SZ) index using the GARCH model. They observed that volatility had a spillover and leveraging impact on these two Chinese stock markets. Siourounis (2002) used GARCH family models to estimate the Athens stock exchange market, revealing that adverse shocks had an asymmetrical influence on everyday returns. Lee (2017) found that asymmetric effects were insignificant in influencing the Korean Composite Stock Price Index (KOSPI). The results of the symmetric and asymmetric GARCH models were nearly identical. Using AR(1)-GARCH (1, 1), Latif et al. (2021) attempted to quantify the impact of COVID-19 on a number of major stock exchanges across the world.

Globalization has caused nations to become more interconnected than in earlier eras, and the influence of globalization can also be seen in the interdependence of country stock markets. According to Glezakos et al. (2007), the Athens stock exchange is dominated by the influence of the United States, while the German and United Kingdom stock markets

also have an impact on the Athens stock exchange. Marjanović et al. (2021) used the Vector Autoregressive model, together with the Granger casualty test and impulse response function analysis, to determine the interrelationship between the stock markets of the United States, Germany, and Japan. They discovered that disturbances in the United States and Germany are transmitted to the Japanese stock market, but not the other way around. Avouyi-Dovi and Neto (2004) examined the stock markets of the United States, France, and Germany to determine the possibility of interdependence. The study discovered that the movement of the French and German stock markets was influenced by the movement of the United States stock market the day before. Nambi (2010) researched the cointegration of the Indian and the U.S. stock markets between 2000 and 2008, finding no correlation between the two markets. However, Das and Bhattacharjee (2020) discovered that NASDAQ affected the Indian stock market from 2010 to 2018. These studies show that there can be spillovers due to foreign markets on the asymmetric risk of a country's stock market.

There have been many studies performed using GARCH models to forecast the volatility. The efficiency of GARCH models in volatility forecasting on the Hong Kong stock market was examined by Liu and Morley (2009) and Wong and Fung (2001) and it was revealed that the GARCH model performs adequately. When it comes to pricing HSI options, Duan and Zhang (2001) discovered that GARCH models beat Black–Scholes models comfortably, while Chen (1997) discovered that the auto-regressive integrated moving average (ARIMA) model performed better than both the mean reversion and the GARCH models in forecasting monthly volatility of the Standard and Poor's 500 index. Vasudevan and Vetrivel (2016) utilized GARCH models to forecast the volatility of the Bombay Stock Exchange–Stock Exchange Sensitive Index (BSE–SENSEX), and discovered that the asymmetric GARCH model performs better than the symmetric GARCH model in forecasting the conditional variance of the BSE–SENSEX return, confirming the existence of leverage effect. Whereas, Banerjee and Sarkar (2006) and Palamalai (2015) found that the symmetric GARCH model outperforms the parsimonious symmetric GARCH model in predicting the volatility returns in the Indian stock market. Palamalai (2011) concluded that the symmetric GARCH model predicts the conditional variance of the S&P 500 index better than the asymmetric one. According to Awartani and Corradi (2005), asymmetric GARCH models outperform the GARCH (1, 1) model. However, asymmetric GARCH models could not beat the other GARCH models with symmetric properties; however, it was concluded that asymmetric GARCH models beat the GARCH (1, 1) model in predicting the volatility of the S&P 500 index. Lim and Sek (2013) utilized GARCH models to capture the volatility of the Malaysian stock market and discovered that symmetric GARCH models perform better than asymmetric GARCH models during the normal period (pre and post-catastrophic), but asymmetric GARCH models perform better during the fluctuation period (catastrophic period). In their analysis of the S&P 100 Index, Liu and Hung (2010) demonstrated that asymmetric GARCH models are more accurate in predicting stock market volatility when there is asymmetric information with different distributions in their investigation of the S&P 100 Index. Lin (2018) utilized GARCH family models to understand the fluctuations of the SSE index and found that the index has a large degree of leptokurtosis along with the presence of ARCH and GARCH effects; he also discovered that the EGARCH (1, 1) model beats the GARCH (1, 1) model and the TARCH (1, 1) model.

Traditional financial time series models are not alone in predicting volatility in the age of big data. Many researchers have now largely focused on utilizing machine learning approaches to estimate stock price indexes. This job is well-suited to deep learning neural networks, such as RNN. Moghar and Hamiche (2020) utilized the LSTM model to forecast major NYSE firms such as Google and Nike and discovered that the LSTM model performed well in predicting these companies' opening and closing values. Qiu et al. (2020) applied the RNN approach to the S&P 500, HSI, and DJI by using a wavelet transform to process stock data to forecast the opening stock prices. They found that LSTM predictors are better when compared with other RNN approaches. Budiharto (2021) utilized LSTM to forecast Indonesian stock prices over a COVID-19 period and discovered that LSTM could uncover

hidden patterns between input and output data for stock price forecasting. Siami-Namin and Namin (2018) examined the accuracy of ARIMA and LSTM, two illustrative methods and discovered that in terms of anticipating the financial data, LSTM beats ARIMA. Shah et al. (2019) conducted a comparison study of RNN and deep neural network (DNN) and concluded that RNN beats DNN in forecasting. Lanbouri and Achchab (2020) studied the forecasting ability of LSTM on the S&P 500 and found it effective in forecasting future prices. Yu and Li (2018) compared the performance of LSTM and GARCH models in predicting the fluctuations of China's equity market. Jung and Choi (2021) used a hybrid model to forecast the volatility of Foreign Exchange (FX) and discovered that the model could accurately predict the volatility; the hybrid model consisted of the combination of autoencoder and LSTM models. In addition, some studies have constructed hybrids of LSTM and GARCH models for predicting the instability in financial assets (Hu et al. 2020; Kim and Won 2018), and in terms of volatility prediction accuracy, empirical data show that hybrid models built using GARCH and ANN techniques perform better. Table 1 shows the previous studies done on various stock markets using quantitative methods.

**Table 1.** Details on quantitative methods utilized by researchers in previous studies.

| Authors | Models Used in the Research | Outcome |
|---|---|---|
| Khan et al. (2019) | GJR-GARCH, GARCH, and EGARCH | GJR-GARCH and EGARCH outperformed GARCH |
| Song et al. (1998) | GARCH (1, 1) and ARMA | GARCH (1, 1) outperformed ARMA |
| Siourounis (2002) | GARCH (1, 1), LGARCH (1, 1), and EGARCH-M (1, 1) | All the models performed satisfactorily |
| Lee (2017) | GARCH (1, 1), GJR-GARCH (1, 1), and QGARCH (1, 1) | No single model outperformed |
| Liu and Morley (2009) | Moving average, GARCH, EGARCH, TGARCH, C-GARCH, and AC-GARCH | EGARCH gave more accurate results |
| Wong and Fung (2001) | GARCH and OLS regression | All the models performed satisfactorily |
| Duan and Zhang (2001) | GARCH and Black–Scholes model | GARCH model performed better than Black–Scholes model |
| Chen (1997) | GARCH, ARMA, and mean reversion | ARMA outperformed all the other models |
| Vasudevan and Vetrivel (2016) | GARCH, EGARCH, and TGARCH | EGARCH and TGARCH performed better than GARCH |
| Palamalai (2011) | GARCH (1, 1), EGARCH (1, 1), and TGARCH (1, 1) | GARCH (1, 1) was better than EGARCH (1, 1) and TGARCH (1, 1) |
| Liu and Hung (2010) | GARCH, EGARCH, and TGARCH | TGARCH and EGARCH outperformed GARCH |
| Lin (2018) | GARCH (1, 1), EGARCH (1, 1), and TARCH (1, 1) | EGARCH (1, 1) outperformed GARCH (1, 1) and TARCH (1, 1) |
| Moghar and Hamiche (2020) | LSTM | LSTM performed satisfactorily in forecasting task |
| Qiu et al. (2020) | LSTM, LSTM with wavelet denoising, and gated recurrent unit | LSTM was better than all the models |
| Siami-Namin and Namin (2018) | LSTM, ARIMA | LSTM outperformed ARIMA |
| Shah et al. (2019) | LSTM, deep neural network | LSTM beat deep neural network |
| Lanbouri and Achchab (2020) | LSTM | LSTM was effective in forecasting the future prices |
| Hu et al. (2020) | LSTM, GARCH, ANN, and hybrid of LSTM–ANN–GARCH | Hybrid model was found to be better than all other models |

### 3. Data Description and Methodology

Figure 1 displays the work flow of the methodology that is applied in this research paper.

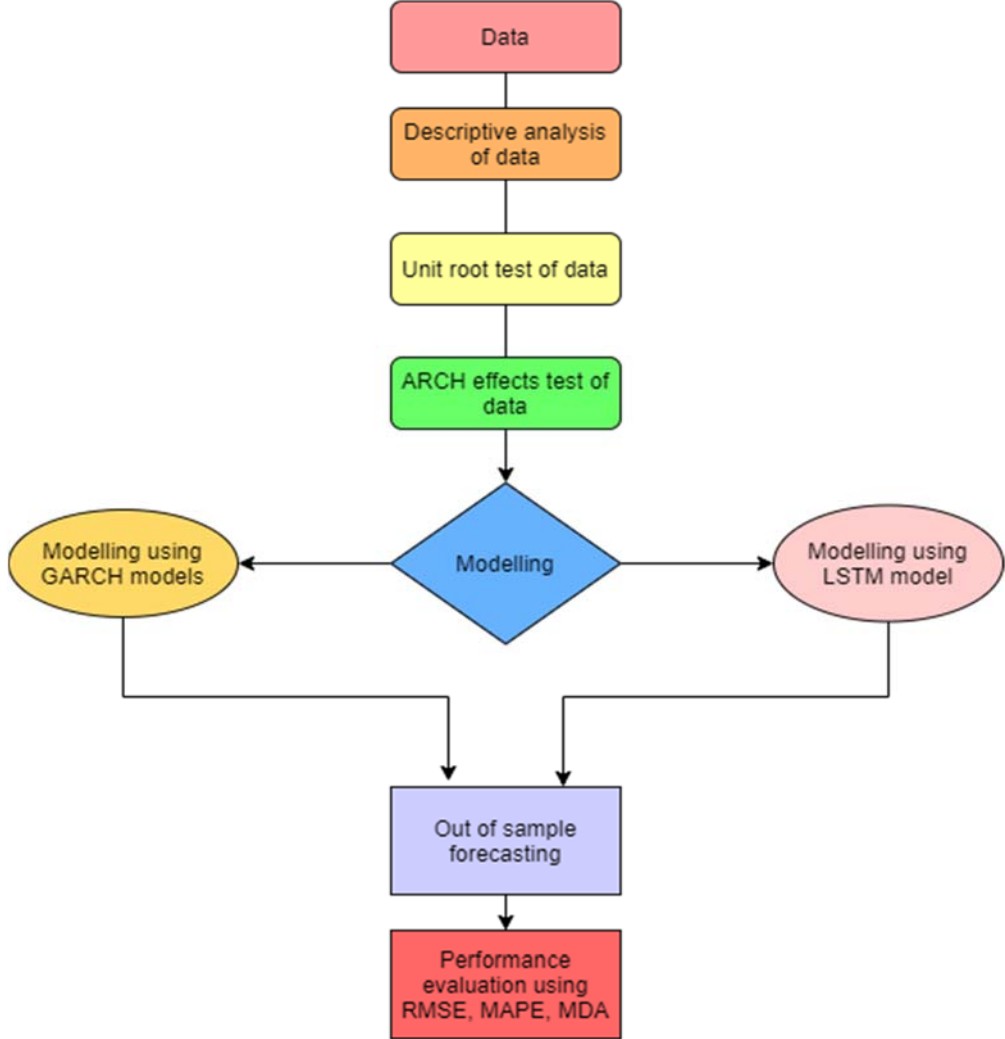

**Figure 1.** Flow chart of the methodology applied in this study.

*3.1. Data Description*

This study uses daily time series data of India Volatility Index (India VIX) data to forecast the volatility of the NIFTY 50 index. India VIX is a volatility index that is based on the NIFTY 50 index option prices. India VIX value (in %) is computed from the best bid-ask prices of the NIFTY 50 index options contracts, indicating the predicted NIFTY 50 index fluctuations in the following thirty calendar days. The India VIX employs the calculation technique used by the Chicago Board Options Exchange (CBOE), but with some modifications to fit the NIFTY options order book. India VIX data collected for this study lies in the period from 1 April 2011 to 26 April 2021, where the last 15 days of data are used for the performance comparison of models using out-of-sample testing. The time duration of 10 years and 15 days was chosen so that models could be trained in a generalized fashion and model comparisons could be conducted over a longer period of time. The information was collected from the Moneycontrol website, which is accessible to everyone for free.

The returns of India VIX used in this paper are of compounded nature:

$$Z_t = \ln\left(D_t / D_{t-1}\right)$$

where $Z_t$ is the return on $t$th day, $D_t$ is the closing data of India VIX on an $t$th day, and $D_{t-1}$ is the closing data of India VIX on $t - 1$th day.

### 3.2. Methodology

This paper investigates the volatility of the National Stock Exchange with the help of econometric and RNN methods. Econometric methods used in this study are GARCH (1, 1), TARCH, and EGARCH, whereas the RNN category is LSTM. Asymmetric techniques like TARCH and EGARCH are used to capture any asymmetry in the India VIX returns. Historical volatility presupposes that the future will be a continuation of the past, with no significant changes. As a result, we can make fair forecasts about future fluctuations based on historical and current values. However, because of various disturbances induced by government policies, the economic conditions, and the psychology of investors; stock volatility exhibits a complex shift at high frequencies which makes it difficult to predict accurately. Still, researchers are developing new methods, which show the potential of forecasting the volatility of stock markets. To represent stock volatility, we may use the following abstract mathematical expression:

$$Z_t = f (Z_{t-1}, Z_{t-2}, \dots, Z_{t-n}),$$

$Z_t$ is the volatility return at period $t$, $Z_{t-1}, Z_{t-2}, \dots, Z_{t-n}$ represents the volatility returns of $n$ periods before the $t$th period, and $f$ denotes the linear or non-linear functional relationship between input and output.

### 3.2.1. Symmetric GARCH Model

The basic concept of the GARCH model is that the residual of its regression model is based on the square error of the preceding data. Additionally, the residual is not independent, despite that it is normally distributed. The conditional variance is the primary focus of attention, and it is used as important information for future variance. The basic GARCH model is GARCH (1, 1), whose equation is as follows:

$$Z_t = \alpha_0 + (\alpha_1 * Z_{t-1}) + E_t, \quad \text{(mean equation)}$$

$$E_t = V_t * K_t,$$

$$V_t{}^2 = \beta_0 + (\beta_1 * E_{t-1}{}^2) + (\gamma * V_{t-1}{}^2), \quad \text{(conditional variance equation)}$$

where $Z_t$ represents the volatility return at period $t$, $K_t$ represents the standardized residual returns (i.e., iid random variable with zero mean and variance 1), $E_{t-1}$ represents the lag of residual values, $V_t$ represents the conditional variance at an instant $t$, $V_{t-1}$ represents the conditional variance at an instant $t - 1$, and $\alpha_0$, $\alpha_1$, $\beta_0$, $\beta_1$, and $\gamma$ are the constant values.

### 3.2.2. Asymmetric GARCH Model

The stock market tends to fall down but then rise again. On the other hand, uptrends tend to be slower and less steep, whereas downtrends are faster and sharper and can result in cascade drops. This is known as the asymmetric volatility phenomenon (AVP), and this phenomenon is often found in financial markets. TARCH Zakoian (1994) and EGARCH Nelson (1991) are mainly used to identify the asymmetric shock in the equity market returns.

#### EGARCH Model

According to the linear GARCH model, equal values favorable (positive) and unfavorable (negative) shocks produce the same amount of fluctuations in the price of equity, i.e., the conditional variances are the same. However, in practice, favorable and unfavorable shocks with similar absolute magnitude can generate varying degrees of volatility, particularly in financial markets. As a result, the asymmetry effect on equity market fluctuation is beyond the explanatory capability of linear GARCH models. Therefore, based on the

GARCH model, Nelson (1991) introduced the exponential GARCH model, often known as the EGARCH model. The governing equation of the EGARCH model is shown below:

$$Z_t = \alpha_0 + (\alpha_1 * Z_{t-1}) + E_t, \quad \text{(mean equation)}$$

$$\ln(V_t^2) = \beta_0 + \sum_{i=1}^{q} \left( \beta_i \left| \frac{E_{t-1}}{V_{t-1}} \right| + \mu_i \left( \frac{E_{t-1}}{V_{t-1}} \right) \right) + \sum_{i=1}^{p} \left( \gamma_i * \ln\left(V_{t-1}^2\right) \right), \quad \text{(conditional variance equation)}$$

where $\ln(V_t^2)$ is the logarithm of conditional variance, $i$ is the order of lagged variable, $\gamma_i$ is the constant value for $i$th order of lag, $\mu_i$ is a coefficient for the lag variable, and $\mu$ is generally used to check the leverage effect in the stock market data modeling. If $\mu_1 = \mu_2 = \ldots = 0$ is true, then the equity price's response to news influence is asymmetric; if $\mu_i < 0$ is true, then asymmetricity exists, and the influence of negative news on the market is more significant than the impact of positive news; and if $\mu_i > 0$ is true, then asymmetric effect exists, but the impact of unfavorable news is weaker than favorable news.

TARCH Model

The threshold GARCH model was created to spot the leverage impact in money-related markets. To do so, a multiplicative dummy term was inducted into the variance equation to see a statistically significant difference during adverse shocks. It is also based on the idea that unforeseen information shocks might impact stock return volatility. The mathematical expressions used in developing the model are as follows:

$$Z_t = \alpha_0 + (\alpha_1 * Z_{t-1}) + E_t, \quad \text{(mean equation)}$$

$$V_t^2 = \beta_0 + \sum_{i=1}^{q} \left( \beta_i * E_{t-1}^2 \right) + \left( \mu * E_{t-1}^2 * K_{t-1} \right) + \sum_{i=1}^{p} \left( \gamma_i * V_{t-1}^2 \right), \quad \text{(conditional variance equation)}$$

If $E_t > 0$, $K_t = 1$; else $K_t = 0$. The coefficient $\mu$ tells the influence of favorable shock, and $\mu + \beta$ tells the influence of unfavorable shock. As a result, if $\beta$ is larger than 0, the negative news impact is greater than the positive news, resulting in an asymmetric effect. There would be no leverage effect if $\beta$ is equal to 0.

3.2.3. LSTM Model

The goal of this research is to use LSTM as a recurrent layer in an RNN model. The data fed into a layer of LSTM is in the form of a vector which is a combination of the input vector xt and the preceding time-step output vector $h_{t-1}$, with ht denoting the output vector from that layer at time t. The LSTM layer comprises four blocks: three gates and one memory cell state $C_t$ (where all of the computations take place). The gates (forget and input gates) perform crucial decisions such as which data needs to be omitted and which data needs to be kept for further training. The output gate also decides the portion of the memory state that needs to be shown in the output. The LSTM contains four computational blocks, which are also termed computation blocks. Vector multiplication is the most important computational task in training a LSTM model. Each block is made up of a matrix which is the result of vector multiplication followed by addition with vector containing bias values. Finally, an activation function is applied to the resulting value. Each block may also include element-by-element multiplication operations. Tanh and sigmoid functions are utilized as activation functions in the LSTM. The following are the four calculation blocks:

**Forget Gate:** The forget gate's job is to determine which data should be forgotten. The output $f_t$ of the forget gate is computed as follows:

$$f_t = \sigma(W_f [h_{t-1}, x_t] + b_f),$$

where $x_t$ represents the input values in form of a vector, $h_{t-1}$ represents the output vector of the hidden state, $W_f$ represents the matrix containing weight, $b_f$ denotes the vector having bias values, and $\sigma$ denotes the sigmoid function.

**Input Gate:** The input gate's role is to figure out what data needs to be updated. The output of the input gate $i_t$ is calculated in the same way as the output of the forget gate.

$$i_t = \sigma(W_i\,[h_{t-1}, x_t] + b_i),$$

$W_i$ is the weight matrix and the $b_i$ represents bias vector.

**State Computation:** The goal of this calculation is to figure out what the new memory state $C_t$ of the LSTM cell is. It calculates the potential values of the new state using the following method:

$$\tilde{C}_t = \tanh(W_c[h_{t-1}, x_t] + b_c),$$

where $x_t$ represents the input in form of a vector, $h_{t-1}$ represents the output of the hidden state in form of a vector, $W_c$ depicts the weight matrix, and $b_c$ depicts the bias values that are put together in a vector. The vector $C_t$ of the new state is then computed using element-wise product of the previous state vector $C_{t-1}$ with the forget gate output vector $f_t$ and the new state candidate vector $\tilde{C}_t$ element-wise with the output vector of the input gate as follows:

$$C_t = f_t \odot C_{t-1} + i_t \odot \tilde{C}_t,$$

where $\odot$ depicts the element-wise multiplication.

**Output Gate:** The main purpose of output gate is to determine the LSTM output. To begin, the vector $o_t$ of the output gate is calculated as follows:

$$o_t = \sigma(W_o[h_{t-1}, x_t] + b_o),$$

where $x_t$ denotes the vector of input, $h_{t-1}$ denotes the output of the hidden state in the form of a vector, $W_o$ represents the weight matrix, $b_o$ represents the vector of bias values, and $\sigma$ depicts the sigmoid function. The invisible state output $h_t$ is then determined through element-wise multiplication of the tanh of state vector $C_t$ with the vector of output gate $o_t$.

$$h_t = o_t \odot \tanh(C_t).$$

## 4. Empirical Analysis

### 4.1. Descriptive Statistics of India VIX Returns Series

The necessary task before processing the data is to have a fundamental understanding of the data series' statistical characteristics. Table 2 shows an overview of descriptive information for the India VIX returns. The mean, maximum, and lowest values, as well as the Jarque–Bera test, skewness, kurtosis, and standard deviation, are all included.

**Table 2.** Autocorrelation, Q-statistic, and probability values of India VIX returns series.

| Lags | AC | Q-Statistic | Probability |
|---|---|---|---|
| 1 | 0.018 | 7.866 | 0.019 |
| 2 | −0.053 | 10.192 | 0.016 |
| 3 | −0.0306 | 10.226 | 0.036 |
| 4 | −0.003 | 11.906 | 0.036 |
| 5 | 0.026 | 12.645 | 0.049 |
| 6 | −0.017 | 12.774 | 0.077 |
| 7 | −0.007 | 14.487 | 0.069 |
| 8 | 0.026 | 19.463 | 0.021 |
| 9 | 0.044 | 25.382 | 0.004 |
| 10 | 0.048 | 29.559 | 0.001 |

The mean of the India VIX return series is 0.0002 percent, or virtually zero, which is consistent with stock market behavior, as stock market return series are regressive when seen over time. The difference between the greatest and minimum value is 0.911207, indicating that the NSE is exceptionally volatile. The standard deviation, at 5.1572 percent, also

shows high volatility. Moreover, the skewness of the data suggests that India VIX returns are positively skewed, telling that data has a long tail on the right side. Consequently, the value of kurtosis (8.543) is significantly higher than the value of standard average distribution (3), which shows that India VIX data has a "sharp peak." Finally, the Jarque–Bera statistic value (7661.286) is much bigger than the standard average distribution value (5.88); therefore, null hypothesis is rejected. Hence, the return series of Indian VIX does not seems to follow a normal distribution; instead, it follows a skewed distribution with positive skewness.

*4.2. Data Processing*

Stationarity study is a significant part of modeling since the GARCH models expect the data to be stationary, so it becomes vital to have stationary data. The autocorrelation plot of the India VIX returns series is shown in Figure 2. The plot illustrates that there is almost no autocorrelation of the data with its lags, and Table 2 (which shows the values of autocorrelation, Q-statistics, and probability) also shows that there is no autocorrelation since the minimum Q-statistics value (7.88) is greater than the 5% significance level (7.81). Figure 3 shows the autocorrelation plot of VIX data using the lag values.

Table 2 confirms that India VIX returns series is stationary, but it may possess white noise, which is still unfit for GARCH models. Hence, taking the first order would be beneficial in removing the possible white noise. Therefore, the series "$\Delta VIX_t$," which is the first-order difference of the India VIX returns in a sequence, is used further in this study.

The results reveal that the decay rate of the autocorrelation function of series $\Delta VIX_t$ is quite fast, approaching zero at period 3. As a result, series $\Delta VIX_t$ can be considered stationary and free of white noise at first glance. This study further utilizes a unit root test to confirm the stationarity of $\Delta VIX_t$.

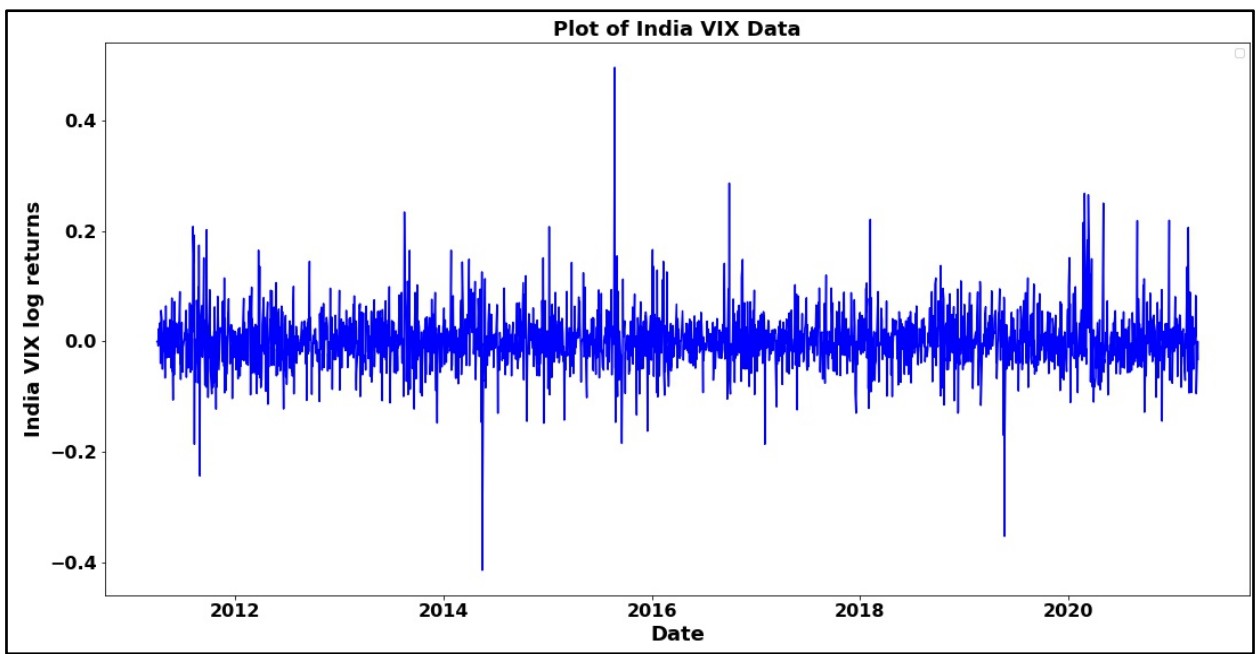

**Figure 2.** Line plot of daily returns of India VIX.

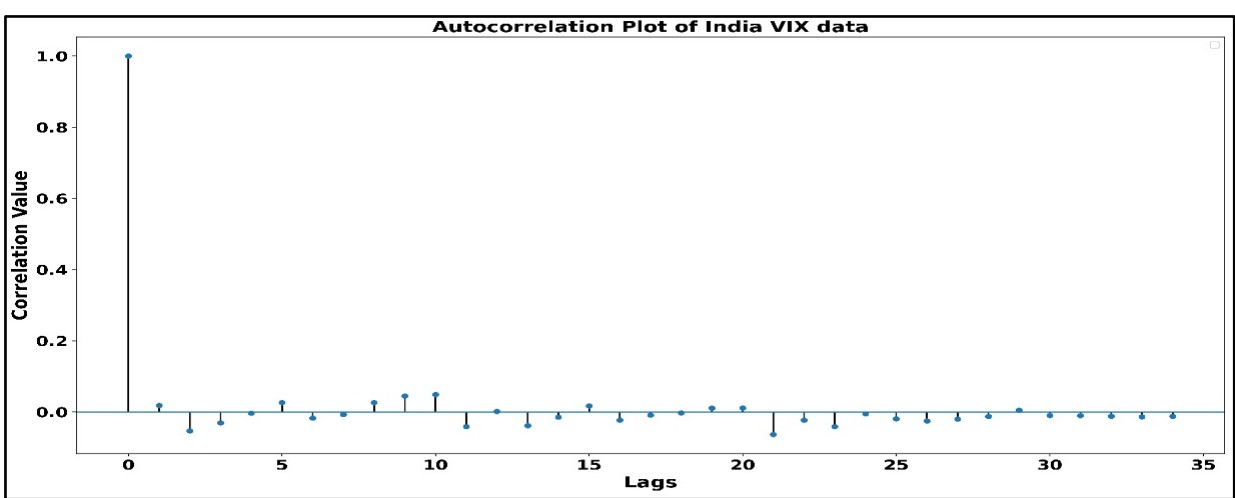

**Figure 3.** Autocorrelation plot of India VIX returns.

### 4.3. Unit Root Test

The Augmented Dickey–Fuller (ADF) test, devised by two American statisticians (Dickey and Fuller 1979), assesses if the autocorrelation coefficient is equal to 1 and is the most widely used approach to test if the series contains a unit root or not. The results of the ADF test were utilized on the $\Delta VIX_t$ series. The series is stationary since the *p*-value is zero, which suggests that null hypothesis should not be accepted; hence, this is evident that $\Delta VIX_t$ is stationary. Furthermore, the $\Delta VIX_t$ series is strongly stationary because the ADF value ($-15.264$) is much smaller than the 1% significance value ($-3.433$).

There is a possibility that $\Delta VIX_t$ may contain structural breaks which lead to non-stationarity so, the Zivot–Andrews unit root test (Zivot and Andrews 1992) was performed to check such a possibility. The *p*-value came out to be 0.00001 which suggests rejecting the null hypothesis (null hypothesis advocates for non-stationarity) hence, $\Delta VIX_t$ is stationary as well as free from structural breaks. Moreover, the Zivot–Andrews statistic was found to be $-13.694$ which is less than 1% significance value ($-5.27644$); therefore, $\Delta VIX_t$ is strongly stationary with almost no structural breaks.

### 4.4. Testing the ARCH Effects

Before using the data for GARCH modeling, it is necessary to check the heteroscedasticity in the data series because GARCH models are developed to capture the fluctuations of heteroscedastic data, which can be achieved by doing the ARCH effects test since ARCH effects are found only in heteroscedastic data. In the pattern of the ARCH effect, first, a big fluctuation occurs which then prompts another significant fluctuation; the ARCH effect also follows another pattern in which a tiny fluctuation is usually followed by another even more small fluctuation. This study employs the ARCH-LM test to check the ARCH effects in the $\Delta VIX_t$ series; Robert Fry Engle III introduced the ARCH-LM test in 1980. The combination of resulting values from the F statistic and the chi-square statistic is primarily used in the process of the ARCH-LM test in order to check the presence of the ARCH effect. The null hypothesis of the ARCH-LM test suggests that the data series lacks ARCH effects; the null hypothesis should be ignored if the F statistic or chi-square statistic is less than 5% critical value; and the null hypothesis needs to be considered if the F statistic or chi-square statistic is greater than 5% critical value.

#### 4.4.1. Selection of Optimum Lag Order for $\Delta VIX_t$ Series

Determination of lag order is a crucial step in time series modeling; this study utilizes Akaike information criterion (AIC), Bayesian information criterion (BIC), and log-likelihood values to find the optimum lag order for the $\Delta VIX_t$ series. Table 3 depicts the AIC, BIC, and log-likelihood values for different lag values of the $\Delta VIX_t$ series, and the best lag order

should have a low value of AIC as well as BIC, whereas the log-likelihood value needs to be high for suitable lag order.

**Table 3.** Autocorrelation, Q-statistic, and probability values of $\Delta VIX_t$.

| Lags | AC | Q-Statistics | Probability |
|------|------|------|------|
| 1 | −0.463 | 538.233 | |
| 2 | −0.047 | 538.248 | |
| 3 | −0.002 | 538.251 | |
| 4 | −0.003 | 541.714 | |
| 5 | −0.001 | 543.571 | |
| 6 | 0.037 | 543.921 | 0.000 |
| 7 | −0.273 | 544.063 | |
| 8 | −0.011 | 544.197 | |
| 9 | 0.007 | 549.873 | |
| 10 | 0.047 | 569.160 | |

From Table 4, it is evident that the most optimum lag order is four because the rate of decrement of AIC and BIC reduces significantly after the lag order. Moreover, the rate of increment of log-likelihood reduces heavily after the 4th lag order. Hence, for time series modeling, the order of lag used in this study is 4.

**Table 4.** AIC, BIC, and log-likelihood values of different lag orders.

| Lags | AIC | BIC | Log-Likelihood |
|------|------|------|------|
| 1 | −5.494 | −5.487 | 3287.659 |
| 2 | −5.612 | −5.602 | 3432.464 |
| 3 | −5.685 | −5.673 | 3522.912 |
| 4 | −5.740 | −5.726 | 3590.704 |
| 5 | −5.763 | −5.747 | 3618.493 |
| 6 | −5.784 | −5.765 | 3643.889 |

4.4.2. Residual Autocorrelation Test to Find the Optimum Lag Order for the ARCH-LM Test

The order of lag is one the most important tasks prior to the ARCH-LM test. This task can be accomplished by utilizing the ARCH (4) model to perform an autocorrelation test on the squared residuals of the $\Delta VIX_t$ series. The value of autocorrelation, Q statistic, and the corresponding probability values are listed in Table 5 below.

**Table 5.** Autocorrelation, Q statistic, and probability values for residuals of the ARCH (4) model.

| Lags | AC | Q-Statistics * |
|------|------|------|
| 1 | −0.4634 | 538.2337 |
| 2 | −0.0479 | 538.2484 |
| 3 | −0.0024 | 538.2517 |
| 4 | −0.0011 | 541.7145 |
| 5 | 0.0373 | 543.5712 |
| 6 | −0.0273 | 543.9217 |
| 7 | −0.0118 | 544.0634 |
| 8 | 0.0073 | 544.1978 |
| 9 | 0.0477 | 549.8739 |
| 10 | −0.0673 | 569.1601 |

* Probability 0.0000.

Autocorrelation values in Table 5 appear to be truncated at lag order 4, with *p*-value being significant at 1% critical level. Hence, a lag order of 4 is most suitable for performing the ARCH-LM test.

### 4.4.3. Performing ARCH-LM Test on $\Delta VIX_t$ Series

The ARCH-LM test is run on residuals generated from the ARCH (4) model. The lag order of residuals utilized in the ARCH-LM is four since autocorrelation values appear to approach zero around lag order 4. The result of the ARCH-LM test on the $\Delta VIX_t$ series is shown in Table 5. The Lagrange multiplier 330.02998, *p*-value (0.00), suggests rejecting the null hypothesis with 1% significance; hence, the $\Delta VIX_t$ series contains the ARCH effect, and the GARCH models can be applied to the $\Delta VIX_t$ series.

### 4.5. Analysis of Results from GARCH (1, 1) Model

Table 6 contains the resultant statistical parameters for the GARCH (1, 1) model when applied on the $\Delta VIX_t$ series with lag order 4.

**Table 6.** Results of the GARCH (1, 1) model.

| Particulars | Value | Particulars | Value |
|---|---|---|---|
| Mean model | | | |
| Variable | Coefficient | t-statistic | *p*-value |
| Constant | 0.001 | 1.194 | 0.233 |
| 1st lag order | −0.781 | −33.646 | 0.000 |
| 2nd lag order | −0.608 | −22.397 | 0.000 |
| 3rd lag order | −0.414 | −14.318 | 0.000 |
| 4th lag order | −0.244 | −10.885 | 0.000 |
| Variance model | | | |
| Omega (constant) | 0.000 | 4.394 | 0.000 |
| Alpha (arch term) | 0.136 | 3.479 | 0.000 |
| Beta (garch term) | 0.707 | 11.716 | 0.000 |
| R-squared | 0.3870 | Adj. R-squared | 0.386 |
| Log-likelihood | 3699.44 | | |
| AIC | −7382.87 | BIC | −7336.37 |

The constant in the mean equation fails the *t*-test, indicating that its value is not significant and thus can be considered zero. However, all of the lag orders' coefficients passed the *t*-test (*p*-value significant at 1%), indicating that their value is substantial and cannot be substituted by zero. In the variance equation, all of the coefficients have passed the *t*-test. The significance found in the alpha (ARCH term) coefficient indicates that the $\Delta VIX_t$ series has asymmetric behavior. In contrast, the importance of the beta (GARCH term) coefficient suggests that the asymmetric behavior is persistent. As a result, it can be inferred that NIFTY 50 volatility is asymmetric throughout the sample period. Furthermore, the R-squared (0.387) and adjusted R-squared (0.386) values demonstrate that the fitting impact is barely acceptable when seen as a complete model. Positive value beta (GARCH term) indicates that the NIFTY 50 volatility returns have a positive risk premium.

### 4.6. Analysis of Results from the EGARCH (1, 1) Model

The results obtained after applying the $\Delta VIX_t$ series with lag order 4 using the EGARCH (1, 1) model are presented in Table 7.

The results show that all the coefficients of the mean model and the variance model passed the *t*-test, proving the coefficients to be significant. An increase in log-likelihood and decrease in AIC and BIC shows that the EGARCH (1, 1) model has performed better than the GARCH (1, 1) model, although the difference is marginal. However, R-squared and adjusted R-squared values favor GARCH (1, 1), and the margin here is also minimal. Moreover, results of the EGARCH (1, 1) model also show that NIFTY 50 contains a leverage effect (unfavorable news is creating greater volatility compared with favorable news) because, during the occurrence of unfavorable news (i.e., Omega less than zero), volatility is increasing (i.e., Gamma greater than zero).

**Table 7.** Results of the EGARCH (1, 1) model.

| Particulars | Value | Particulars | Value |
|---|---|---|---|
| | Mean model | | |
| Variable | Coefficient | t-statistic | *p*-value |
| Constant | 0.002 | 2.371 | 0.017 |
| 1st lag order | −0.765 | −66.993 | 0.000 |
| 2nd lag order | −0.597 | −30.826 | 0.000 |
| 3rd lag order | −0.403 | −17.525 | 0.000 |
| 4th lag order | −0.248 | −14.838 | 0.000 |
| | Variance model | | |
| Omega (constant) | −0.699 | −3.396 | 0.000 |
| Alpha (arch term) | 0.232 | 3.899 | 0.000 |
| Gamma (leverage term) | 0.098 | 2.851 | 0.004 |
| Beta (garch term) | 0.877 | 24.756 | 0.000 |
| R-squared | 0.386 | Adj. R-squared | 0.385 |
| Log-likelihood | 3709.450 | | |
| AIC | −7400.900 | BIC | −7348.580 |

### 4.7. Analysis of Results from the TARCH (1, 1) Model

Table 8 shows the TARCH (1, 1) model results used on the $\Delta VIX_t$ series with lag order 4.

**Table 8.** Results of the TARCH (1, 1) model.

| Particulars | Value | Particulars | Value |
|---|---|---|---|
| | Mean model | | |
| Variable | Coefficient | t-statistic | *p*-value |
| Constant | 0.002 | 3.335 | 0.000 |
| 1st lag order | −0.766 | −41.281 | 0.000 |
| 2nd lag order | −0.599 | −44.553 | 0.000 |
| 3rd lag order | −0.407 | −20.015 | 0.000 |
| 4th lag order | −0.250 | −17.305 | 0.000 |
| | Variance model | | |
| Omega (constant) | 0.007 | 3.138 | 0.001 |
| Alpha (arch term) | 0.191 | 4.556 | 0.000 |
| Gamma (leverage term) | −0.116 | −2.786 | 0.005 |
| Beta (garch term) | 0.776 | 12.188 | 0.000 |
| R-squared | 0.386 | Adj. R-squared | 0.385 |
| Log-likelihood | 3708.480 | | |
| AIC | −7398.960 | BIC | −7346.640 |

All the coefficients of the mean equation of the TARCH (1, 1) model were significant. The lag order coefficients' negative values show that the $\Delta VIX_t$ series' previous values are proportional to the next deal. Log-likelihood, AIC, and BIC values of TARCH (1, 1) have slightly decreased compared with the values of EGARCH (1, 1), which gives mixed conclusions since low AIC and BIC values show the model is good whereas low log-likelihood shows the model is imperfect. The gamma coefficient failed to pass the *t*-test for 1% significance which shows that nothing can be determined regarding the leverage effect from the results of this model.

### 4.8. Finding the Architecture of LSTM

This study compares econometrics models such as GARCH, TARCH, and EGARCH with machine learning techniques such as LSTM. It is necessary to develop a good LSTM model, which significantly depends on the number of layers to be used and the number of nodes present in each layer. Since the LSTM model is compared with GARCH models, to have a fair comparison, the lag order of the $\Delta VIX_t$ series for the LSTM model is kept

the same as that of the GARCH models, i.e., 4. By comparing the AIC and BIC values of training and validation data, the number of layers and nodes in each layer of the LSTM was manually determined. We calculated the AIC and BIC values for several layer and node combinations and chose the one that performed the best. ReLU activation function is used between the layers, and the optimizer used to minimize the training error is Adam because Adam has been proven to be the best among all the 1st order optimizers available for neural networks (Kandel et al. 2020). Table 9 depicts the performance of the LSTM model using different combinations of layers and nodes, and the best combination for the $\Delta VIX_t$ series is found to be three LSTM layers with 80, 80, and 40 nodes in each layer, respectively.

**Table 9.** Comparison of different architectures for the LSTM model.

| Structure of LSTM Model | AIC of Train Data | AIC of Validation Data | BIC of Train Data | BIC of Validation Data |
|---|---|---|---|---|
| 5 | −10,673.5 | −3475.3 | −10,651.4 | −3457.6 |
| 10 | −10,678.7 | −3474.1 | −10,656.6 | −3456.4 |
| 20 | −10,681.1 | −3476.6 | −10,659.0 | −3458.9 |
| 40 | −10,698.1 | −3480.6 | −10,676.0 | −3462.9 |
| 80 | −10,697.5 | −3481.6 | −10,675.4 | −3463.9 |
| 160 | −10,548.0 | −3428.0 | −10,525.9 | −3410.3 |
| 80-20 | −10,660.5 | −3473.5 | −10,638.5 | −3455.8 |
| 80-40 | −10,701.9 | −3484.2 | −10,679.9 | −3466.5 |
| 80-80 | −10,723.7 | −3488.5 | −10,701.6 | −3470.8 |
| 80-160 | −10,596.4 | −3447.9 | −10,574.3 | −3430.2 |
| 80-80-20 | −10,729.3 | −3483.0 | −10,707.2 | −3465.3 |
| 80-80-40 | −10,728.4 | −3484.1 | −10,706.3 | −3466.4 |
| 80-80-80 | −10,724.7 | −3481.4 | −10,702.6 | −3463.7 |
| 80-80-40-20 | −9794.6 | −3171.5 | −9772.5 | −3153.8 |

*4.9. Comparison of LSTM and GARCH Models Using out of Sample Forecasting Results*

This study compares the performance of the models by forecasting the India VIX volatility from each model for the next 15 days in the future and then analyzing the results based on three performance metrics, i.e., mean directional accuracy (MDA), mean absolute percentage error (MAPE), and root mean squared error (RMSE). The reason for keeping three performance metrics is to compare the performance of models in different aspects; for example, MDA would tell the performance of models in predicting the direction of volatility, whereas RMSE would explain how far the predictions are from actual values. Table 10 presents the out-of-sample version of models (LSTM and GARCH models) on the MAPE metric in predicting the India VIX values for the next 10 and 15 days.

**Table 10.** Performance comparison using MAPE, RMSE, and MDA.

| Model | Performance on MAPE Metrics | | Performance on RMSE | | Performance on MDA | |
|---|---|---|---|---|---|---|
| | MAPE on 10 Days Data (in %) | MAPE on 15 Days Data (in %) | RMSE on 10 Days Data | RMSE on 15 Days Data | MDA on 10 Days Data | MDA on 15 Days Data |
| GARCH (1, 1) | 9.598 | 19.176 | 2.541 | 5.724 | 0.4444 | 0.4285 |
| EGARCH (1, 1) | 8.615 | 16.992 | 2.234 | 5.057 | 0.4444 | 0.4285 |
| TARCH (1, 1) | 8.609 | 17.029 | 2.233 | 5.073 | 0.4444 | 0.4285 |
| LSTM | 15.594 | 26.901 | 4.002 | 7.514 | 0.4444 | 0.4285 |

4.9.1. Performance Comparison Using MAPE

In statistics, the mean absolute percentage error, also defined as mean absolute percentage deviation, measures a forecasting method's prediction accuracy. MAPE is determined using the formula given below:

$$\text{MAPE} = \frac{1}{n} \sum_{k=1}^{n} \left| \frac{A_k - F_k}{A_k} \right|$$

where MAPE denotes the mean absolute percentage error, $n$ denotes the total number of samples, $A_k$ denotes the actual value for the $k$th sample, and $F_k$ denotes the forecasted value for the $k$th sample.

The findings show that econometrics models (GARCH, EGARCH, and TARCH) beat the LSTM model in all three performance areas (10 days and 15 days). The TARCH model is the best overall performer. Still, EGARCH outperforms TARCH when tested on 15-day data, indicating that EGARCH is better at forecasting far-future data while TARCH is superior at predicting near-future data. The MAPE increases as the amount of out-of-sample data grows, which is reasonable because error propagation becomes more dominant as the amount of out-of-sample data produces. On 15 days of data, the MAPE value of 16.9925 is a good value.

4.9.2. Performance Comparison Using RMSE

The square root of the average squared deviation between the expected and fundamental values derived from a model is the RMSE. The root mean of the squared error is represented below:

$$\text{RMSE} = \sqrt{\frac{\sum_{i=1}^{N}(x_i - \overline{x_i})^2}{N}}$$

where RMSE is the root mean squared error; $i$ is the observation count; $x_i$ is the actual value; $\overline{x_i}$ is the observed value, and $N$ denotes the total count for the observations. The performance of models on the RMSE metric is in line with the performance of models on the MAPE metric. The TARCH model is shown to be superior in overall performance. Still, when evaluated on 15-day data, EGARCH beats TARCH, suggesting that EGARCH is better at forecasting far-future data while TARCH is better at predicting near-future data.

4.9.3. Performance Comparison Using MDA

MDA is a measure of a forecasting method's prediction accuracy. It compares the predicted direction (upward or downward) to the actual direction. MDA formula is described below:

$$\text{MDA} = \frac{1}{N} \sum_t 1_{sign\ (X_t - X_{t-1}) = sign(F_t - X_{t-1})}$$

$X_t$ depicts the actual observations of the time series, $F_t$ depicts the predicted or forecasted time series, $N$ denotes the count for the data points, $sign$ () denotes the sign function, and 1 is the indicator function. MDA values suggest that in predicting the direction of India VIX values, all the models have performed similarly; hence, no conclusion can be driven for distinguishing the performance of the models. However, the MDA values for different samples (10 and 15 days) show that the model's performance is better on ten days data than 15 days, which is theoretically correct due to error propagation. The MDA value of 0.428 on 15 days shows that models have given satisfactory performance.

**5. Discussion**

The study's primary objectives were twofold. The initial task was the utilization of GARCH models to detect anomalies in the volatility of the NIFTY 50 index. The second was to compare the volatility forecasting ability of econometric models with state-of-the-art machine learning approaches. The researchers used ten years of India VIX data (a volatility indicator for the NIFTY 50 index). The study found that the volatility of NIFTY 50 is interestingly heteroscedastic because ARCH effects were visible in India VIX returns. This research also discovered that the volatility of the NIFTY 50 index has asymmetric effects (or volatility clustering) since the leverage effect was apparent in India VIX returns, indicating that bad news is more impactful in the Indian stock market than positive news. Hence, retail investors are advised to invest more cautiously when lousy news is circulating in India.

When it comes to volatility forecasting, EGARCH outperformed two performance metrics (MAPE and RMSE). In contrast, all models performed similarly in the third metric

(DAR), suggesting no apparent victor, but econometric models (GARCH, EGARCH, and TARCH) had a slight advantage. Moreover, asymmetric GARCH models (EGARCH (1, 1) and TARCH (1, 1)) were found to be slightly better than the symmetric GARCH model (GARCH (1, 1)) in forecasting the volatility of the NIFTY 50 index.

In this study, empirical analysis of models shows that asymmetric models, i.e., EGARCH (1, 1) and TARCH (1, 1), performed better than the GARCH (1, 1) model on MAPE and RMSE, whereas the MDA metric showed that all the models performed equal. Therefore, overall, asymmetric models had a slight edge over the symmetric one which is in line with previous research studies such as Lin (2018), Liu and Hung (2010), Liu and Morley (2009), and Vasudevan and Vetrivel (2016). Moreover, our research concludes that all the GARCH models had a slight edge over the RNN-based LSTM, although Hu et al. (2020) and Kim and Won (2018) claim that hybrid models made of LSTM and econometric models perform better than individual models such as GARCH and LSTM. This shows that the LSTM model has potential but more specialized studies are required to generate concrete evidence.

## 6. Conclusions

The leverage effect indicates that investors are intentionally shorting the NIFTY 50 index futures contract and making it difficult for retail investors to sustain in Indian stock markets. Therefore, the Indian government should provide strengths to retail investors and the Securities and Exchange Board of India (SEBI) to form new policies to stop the manipulation of Indian stock markets. The Indian government should also provide knowledge about Indian stock markets so that retail investors can enter the market with the right mindset and sustain stock markets for a more extended period. More research can be conducted in hybrid models by combining the econometric models with machine learning techniques. Researchers can also try to include technical indicators in input data to make more robust volatility forecasting models. For an experiment, opinions of famous personalities are used in input and technical data to make a better volatility forecasting model.

To assess the volatility of the NIFTY 50 index, this study includes the GARCH model family. Through an empirical analysis of the econometric models such as the symmetric GARCH model (GARCH (1, 1)) and the asymmetric GARCH model (EGARCH, TARCH), this work tries to uncover the peculiarities in the volatility of the NIFTY 50 index. This paper focuses on the volatility of the whole National Stock Exchange (NSE) and the NIFTY 50 index, a collection of 50 top equities of NSE that roughly mirrors the NSE's performance. As a result, it was decided to investigate the NIFTY 50 index's volatility. This research aims to see if these new and advanced econometric models can capture the NIFTY 50's volatility spikes. This research aims to analyze the present state of the Indian stock market, and analyzing the NIFTY 50 index is an excellent way to do so because most of India's day trading happens at NSE. This research also aims to assess the RNN-based LSTM algorithm's capability in capturing the volatility of the NIFTY 50 index and comparing its performance to that of GARCH models on out-of-sample data. With the aid of GARCH models, this study also seeks to uncover asymmetric traits present in index volatility.

Because a large portion of money is bet on stock market index movement, it becomes necessary for brokers to provide a solid risk management system for investors, and the models used in this study may be used to create such a system. Moreover, regulations should be enacted to enhance SEBI, giving it the authority to take swift and strong action against market manipulators. To find these market manipulators, SEBI can certainly use our model to keep a watch on market volatility. Finally, NSE should provide relevant information about stock market movement and stock market volatility to all the brokers and investors and NSE must also alarm its service receivers about rough movements that could potentially occur in the stock market where our models can be of assistance.

Finally, to make the system more robust, a research study might be conducted to determine which nations have the potential to affect the Indian stock market. Such a research study would aid in identifying whether or not the spillover effect may be detected

in the downside risk due to influential nations, which will aid investors in making better investment selections.

**Funding:** This paper has received no external funding.

**Institutional Review Board Statement:** Not applicable.

**Informed Consent Statement:** Not applicable.

**Data Availability Statement:** The code and data used to generate the results for this paper are available here: https://github.com/aashishmalik7936/Modelling-and-forecasting-the-volatility-of-NIFTY-50-using-GARCH-and-RNN-models (accessed on 10 May 2021).

**Acknowledgments:** We greatly thank Susham Biswas, Assistant of Geoinformatics at Rajiv Gandhi Institute of Petroleum Technology for sharing his wisdom during the preparation of this research manuscript. We show our gratitude to the all the reviewers for providing valuable suggestions.

**Conflicts of Interest:** The authors declare no conflict of interest.

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
