# Peer review of "Modeling and Forecasting the Volatility of NIFTY 50 Using GARCH and RNN Models"

_economies, doi:10.3390/economies10050102_

Round 1

Author Response

1. The GARCH models applied to finance literature are standard and have been used since the publication of French et al. (1987). GARCH models are mainly based on the ARIMA specification. Some recent developments such as downside risk (Bali et al., 2009), economic policy uncertainty (Baker et al., 2016; Liu and Zhang, 2015.), equity market volatility (Baker et al. 2020) and COVID-19 (Latif et al., 2021) are excluded from the mean equation of Nifty 50 VIX series. Will your model be biased for these exclusions?

Response 1:  We wanted to do a comparison of generalized models and we do not want to include extra for COVID-19 or other factors in any of our models so that model does not get trained for some specialized factor. That is why the period taken for the study was also sufficiently large.

2. The content of research method contains no innovation. Since the publication of a review article by Bollerslev (1992) and the collection by Engle (1995), the GARCH models are welled understood. Bollerslev (2010) provides variants of the GARCH model, which contains alternative models for reference. 

Response 2: This paper may not have a new method or innovation but this study explores a new angle of research on the Indian Stock Market as there is very little research done in the field which our paper investigates.

3. There is no need to discuss the long history of GARCH model development. Yet, the GARCH models are limited to three forms in this study, why not other forms of risk model? Why Fifty 50 is applied in your paper, not others? It would be of interest to expand the sample by including other indices of other countries.

Response 3: There are other forms of GARCH models that exist but all the forms are categorized into two categories i.e. symmetric GARCH models & asymmetric GARCH models, and this study majorly wants to compare symmetric & asymmetric GARCH models so, from each category, the forms were chosen. We tried other forms as well for instance HARCH but its performance was not different from that of other forms so, we decided not to include unnecessary information in the paper.
 This paper used the NIFTY 50 index because we primarily want to study the Indian stock market (as there is a fair amount of research done of other countries' stock market data) and most of the trading of the Indian Stock market happens on NSE (National Stock Exchange) whose major index is NIFTY 50.          It would be interesting to expand for indexes but that is not the goal of this study.

4. Are the results of horse-racing for out-of-sample forecasting by using RMSE and MDA statistically significant?

Response 4: The performance comparison among the GARCH models is not statistically significant on any of the metrics but the performance comparison of any of the GARCH models with the LSTM model is statistically significant on MAPE & RMSE and statistically insignificant on MDA which overall shows that GARCH models have edge over LSTM model statistically. For instance, performance comparison of TARCH (1, 1) with LSTM on MAPE metric shows the difference of 6.985 percentage and, we have kept TYPE- 1 error to be 5%, hence null hypothesis is rejected which means results are statistically significant.

5. I don’t believe the manuscript is ready for submission. The writing is confusing and notations are lack of clarity. The style is unable to keep up with the professional standard. I mentioned a few as follows. 
• In equation (1), Zi = ln (Di / Di-1) represents the volatility return, and later on, Zt is used (both i and t are used); Et-1 represents the lag of residual values, Kt represents the standardized residual returns. ???? is usually used to denote an error series. 
• Defining “FD” as the first order difference of the VIX return is confusing. Why don’t you used the ∆????????? The variable is taking the first difference and with subscript t. Using FD does not identify the series being differenced. 
• For Tables 5-7, the rows of R-Squared, Log-Likelihood, and AIC should be placed on the bottom. 
• In time series model, it is usually report of Ljung-Box Q statistic for test autocorrelation, instead of Durbin-Watson statistic, since the latter only applies to the first order correlation. 
• The way to cite papers is incorrect. For instance, “(Nelson, 1991)” introduced the Exponential GARCH model. It should be written as: Nelson (1991) introduced…. Similar errors apply to other citations in the text.

Response 5: We have done all the changes that were suggested in this query, please review the updated manuscript to find the changes.

6. The following references may be useful to this study. 

Response 6: We have included useful references in the manuscript.

Reviewer 2 Report

Dear Authors,

The four models of GARCH (1, 1), TARCH (1, 1), EGARCH (1, 1), and LSTM are fitted for the daily time-series data of the India Volatility Index (India VIX) in the period from April 1 2011 to April 26, 2021. The last 15 days of data are used for the performance comparison. the results show that three GARCH models are slightly better than the LSTM model in forecasting ability.

However, there are still some points that need to be explained and tested more clearly.

  1. Index data after taking a log and first difference are close to the returns of Index. How can these returns be used to predict the volatility of the index? (First difference of LOG = percentage change). So, is the concept of risk here correct?
  2. Why are GARCH (1,1), TARCH (1, 1), and EGARCH (1,1) models chosen to conduct the test but not GARCH (n,n), TARCH (n, n), or GARCH (n, n)?
  3. For conditional variance models, such as the popular GARCH model, testing the adequacy of the estimated model is necessary before giving relevant applicability in practice. Several misspecification tests do exist for GARCH models as well. For example, Bollerslev (1986) already suggested a score or Lagrange multiplier (LM) test for testing a GARCH model against a higher-order GARCH model. Li and Mak (1994) derived a portmanteau type test for testing the adequacy of a GARCH model. Engle and Ng (1993) considered testing the GARCH specification against asymmetry using the so-called sign-bias and size-bias tests. Chu (1995) derived a test of parameter constancy against a single structural break. This test has a nonstandard asymptotic null distribution, but Chu provided tables for critical values. Lin and Yang (1999) derived another test against a single structural break, based on empirical distribution functions.
  4. The performance of the LSTM model is presented in Table 8 with different combinations of layers & nodes. However, the best LSTM model is not only determined by the number of nodes and layers but also by the number of timesteps, batch size, epochs, etc. How to decide the three LSTM layers with 80, 80, and 40 nodes in each layer is the best combination for the series

References

  • Bollerslev, T., 1986. Generalized autoregressive conditional heteroskedasticity. Journal of Econometrics 31, 307–327.
  • Chu, C.-S.J., 1995. Detecting parameter shift in GARCH models. Econometric Reviews 14, 241–266.
  • distribution function approach. Research Paper 30, University of Technology, Quantitative Finance Research Group, Sydney.
  • Engle, R.F., Ng, V.K., 1993. Measuring and testing the impact of news on volatility. Journal of Finance 48, 1749–1777.
  • Li, W.K., Mak, T.K., 1994. On the squared residual autocorrelations in non-linear time series with conditional heteroskedasticity. Journal of Time Series Analysis 15, 627–636.
  • Lin, S.-J., Yang, J., 1999. Testing shift in financial models with conditional heteroskedasticity: an empirical
  • Nelson, D.B., 1990. Stationarity and persistence in the GARCH (1,1) model. Econometric Theory 6, 318–334.

Author Response

1. Index data after taking a log and first difference are close to the returns of Index. How can these returns be used to predict the volatility of the index? (First difference of LOG = percentage change). So, is the concept of risk here correct?

Response 1: Index data after taking the difference of log of returns is very different from index data with returns. For instance: Let’s call FD series as the first difference of log returns and RET as return then,
Mean of FD: -0.000013                                                        
Standard deviation of FD: 0.07222
Skewness of FD: -0.5069
Kurtosis of FD: 5.6201

Mean of RET: -0.000048                                         
 Standard deviation of RET: 1.20418
Skewness of RET: 0.9863
Kurtosis of RET: 21.9689
All the above statistics suggest that both the series are very different. Finally, the main purpose of doing the first difference was to remove the potential white and the first difference series showed very strong decay to autocorrelation which is healthy for stationarity.

2. Why are GARCH (1,1), TARCH (1, 1), and EGARCH (1,1) models chosen to conduct the test but not GARCH (n,n), TARCH (n, n), or GARCH (n, n)?

Response 2: We utilized GARCH (1, 1), TARCH (1, 1) & EGARCH (1,1) because there have been many studies Gulay & Emec (2019); JAFARI et al. (2007); Namugaya et al. (2014) confirming that GARCH (1, 1) models work best in general. 

References:

Gulay, E., & Emec, H. (2019). The Stock Returns Volatility based on the GARCH (1,1) Model: The Superiority of the Truncated Standard Normal Distribution in Forecasting Volatility. Iranian Economic Review, 23(1), 87–108. https://doi.org/10.22059/ier.2018.69100

JAFARI, G. R., BAHRAMINASAB, A., & NOROUZZADEH, P. (2007). WHY DOES THE STANDARD GARCH(1, 1) MODEL WORK WELL? International Journal of Modern Physics C, 18(07), 1223–1230. https://doi.org/10.1142/S0129183107011261

Namugaya, J., Weke, P. G. O., & Charles, W. M. (2014). Modelling Volatility of Stock Returns: Is GARCH(1,1) enough? International Journal of Sciences: Basic and Applied Research (IJSBAR), 16(2), 216–223. https://gssrr.org/index.php/JournalOfBasicAndApplied/article/view/2483

3. For conditional variance models, such as the popular GARCH model, testing the adequacy of the estimated model is necessary before giving relevant applicability in practice. Several misspecification tests do exist for GARCH models as well. For example, Bollerslev (1986) already suggested a score or Lagrange multiplier (LM) test for testing a GARCH model against a higher-order GARCH model. Li and Mak (1994) derived a portmanteau type test for testing the adequacy of a GARCH model. Engle and Ng (1993) considered testing the GARCH specification against asymmetry using the so-called sign-bias and size-bias tests. Chu (1995) derived a test of parameter constancy against a single structural break. This test has a nonstandard asymptotic null distribution, but Chu provided tables for critical values. Lin and Yang (1999) derived another test against a single structural break, based on empirical distribution functions.

Response 3: We did many preliminary tests before using the data in the GARCH models. For instance, the feeding data for the GARCH model needs to be stationary, without structural breaks, autocorrelation should not be present and it must have ARCH effects so, we did ADF test to check stationarity, Zivot-Andrews test for checking structural breaks, First difference of series showed strong resistance against any autocorrelation, and ARCH-LM test for confirming the presence of ARCH effects.

4. The performance of the LSTM model is presented in Table 8 with different combinations of layers & nodes. However, the best LSTM model is not only determined by the number of nodes and layers but also by the number of timesteps, batch size, epochs, etc. How to decide the three LSTM layers with 80, 80, and 40 nodes in each layer is the best combination for the series

Response 4: We kept epochs to be sufficiently large (1000) and applied early stopping with the condition that training will stop as soon as the error later epoch becomes larger than the former one. So, all the models got sufficient epochs to train themselves.
For batch size, timestamp and dropout we experimented with different values and found that batch size of value 4, timestamp with value 1, and dropout with value 0.0 were working best, the rest of all the hyperparameters were kept default. These all tests were done on a single LSTM layer with 5 nodes. Then, we started experimenting on LSTM layers and nodes and reached 80-80-40.
We agree that the final combination of our hyperparameters may not produce the most efficient model but our hyperparameters definitely produce a very efficient as we have tried to cover most of the permutation possible with the hyperparameters and it becomes a very difficult & time-consuming task to check all the possible permutations with the hyperparameters.

Reviewer 3 Report

Dear Author(s),

Please find attached the Review Report.

Author Response

 1. The study is limited to merely the Indian financial market. In this regard,
a strong motivation for selecting such a market is more than necessary;

Response 1: The motivation behind using only Indian stock market data has been included in the last paragraph of the “Research Contribution” section.

2. The introductory section should emphasize clearly the novelty and originality of the manuscript. There are many other papers that explored the volatility of various stock markets. As such, the contribution of current research to the existing literature is not convincing;

Response 2: The problem of novelty should be solved as we have merged the research contribution into an introduction that will clearly show the novelty and originality of the manuscript.
We agree that many researchers have explored the volatility of various stock markets but for the Indian stock market, the research done in applying machine learning to forecast volatility is very little.
We have added more references as well as a table in the literature review which improves the literature review very much.

3. The first two sections should be merged within a single section. As well, the introductory section requires an outline of the whole manuscript;

Response 3: We did the changes as suggested in this point and the “Introduction” section now serves all the aspects that were suggested.

4. The section related to prior literature discussion should be appended with
a table summarizing the quantitative techniques employed in earlier studies, as well as their outcome;

Response 4: We have added the table in the “Literature Review” section.

5. The period selection (e.g., April 1, 2011, to April 26, 2021) should be argued;

Response 5: This point has been addressed in the “Data Description” section.

6. The COVID-19 pandemic stage should be considered distinctly (e.g., pre-pandemic and post-pandemic-periods); 

Response 6: The aim of this study is to analyze the models in a generalized manner and not a certain event or factor so, considering pre-covid and post-covid is out of scope for this study but it is definitely a topic worth exploring for research.

7. The unit root investigation should take into account possible structural breaks;

Response 7: The test to detect structural breaks was conducted and the description has been added in the “Unit Root Test” section.

8. The EWMA (Exponentially Weighted Moving Average) should be also considered. Different to GARCH models, EWMA has the benefit of a non-return to average;

Response 8: This study mainly focuses on investigating GARCH models & RNN models, and the use of EWMA is not the scope of this research paper but EWMA can be explored in other research studies as it will be interesting to see how EWMA performs compared to other models.

9. In addition to GARCH (1, 1), EGARCH (1, 1) and TARCH (1, 1), the author(s) should also consider PGARCH and IGARCH;

Response 9: We made PGARCH and IGARCH models using the data we decided not to include them because:
   a. Results of these two models were not statistically different from EGARCH and TARCH models.
   b. PGARCH and IGARCH models were not providing any new information that would be useful for our research.
   c. Main focus of this study was to compare RNN models with symmetric and asymmetric GARCH models, and we have already selected the GARCH models for each category.

10. Apart from technical discussions, the empirical outcomes should be interpreted and compared with earlier literature; 

Response 10: We have added the comparison in the last paragraph of the “Discussion” section.

11. The policy and practical implications are very weak. Hence, an extended
discussion is necessary. 

Response 11: More practical and policy implications have been added in the  “Conclusion” section.

Round 2

Reviewer 1 Report

  1. The U.S. influence is needed to enhance.
  2. The downside risk may spillover from global market (Chen, et al. 2018, JBF).

Author Response

1. The U.S. influence is needed to enhance.

Response: We have increased the USA influence by adding a paragraph (i.e. 4th paragraph on the new manuscript) in the “Literature Review” section. 

2. The downside risk may spillover from global market (Chen, et al. 2018, JBF).

Response: This research is primarily focused on developing better volatility forecasting methods for Indian stock market data, but downside risk spillover from other countries' stock markets also affects volatility, necessitating a thorough comparison of the Indian stock market with other markets, which is beyond the scope of this research. However, in the "Conclusion" section, we have added a paragraph (the last paragraph) that discusses potential spillovers from global markets, since the comparative analysis will assist investors and the Indian stock market watchdog in making better decisions.

Reviewer 2 Report

Dear Authors,

1. In finance, there are many types of rate of returns  (https://en.wikipedia.org/wiki/Rate_of_return), using the appropriate type is fine. The problem here is "Rate of return of Volatility".

India VIX or India Volatility Index is a volatile index that is calculated by the NSE to measure the market's anticipation for volatility and fluctuations in the near term. This shows the increase/fall in volatility in the market (not the return as explained in the manuscript). This index represents the investors' perception of the market over the next near term, that is the next 30 days. The rise and fall in the India VIX or volatile index determines the volatility of the market and helps the investors to better understand the market conditions before making their next big investment or while keeping a track of their previously made investment. It is important to know that the volatile index is in no way similar to the price index like the NIFTY. While the price index is calculated by taking into account the price movement of the underlying equities, the volatile index or India VIX is calculated using the orderbook of the underlying index options and is represented in the form of a percentage. So what is the point of this whole study?

2. One model can be good for fitting some data, but it is not meant to be suitable for all data. Therefore, before drawing certain conclusions, it is advisable to prove that the applied model is suitable through robustness or diagnostic tests. If the model used is not tested with certainty, it will lead to biased conclusions.

Author Response

1. In finance, there are many types of rate of returns  (https://en.wikipedia.org/wiki/Rate_of_return), using the appropriate type is fine. The problem here is "Rate of return of Volatility".

India VIX or India Volatility Index is a volatile index that is calculated by the NSE to measure the market's anticipation for volatility and fluctuations in the near term. This shows the increase/fall in volatility in the market (not the return as explained in the manuscript). This index represents the investors' perception of the market over the next near term, that is the next 30 days. The rise and fall in the India VIX or volatile index determines the volatility of the market and helps the investors to better understand the market conditions before making their next big investment or while keeping a track of their previously made investment. It is important to know that the volatile index is in no way similar to the price index like the NIFTY. While the price index is calculated by taking into account the price movement of the underlying equities, the volatile index or India VIX is calculated using the orderbook of the underlying index options and is represented in the form of a percentage. So what is the point of this whole study?

Response: We agree that India VIX is generated based on NIFTY 50 options order books, but NIFTY 50 index pricing is dependent on stock prices, yet research studies have revealed a statistically significant association between NIFTY 50 and India VIX Bantwa (2017). Furthermore, Just & Echaust (2020); Ozair (2014); Magner et al. (2021) discovered that there is a strong relationship between CBOE VIX and S&P500, and the underlying method used in calculating India VIX is the same as that used by CBOE, implying that the VIX and stock market index have a statistically significant relationship.

Finally, to assess the performance of the models, the rate of return projected by each model is utilized in back-calculation for each data point to get the anticipated India VIX value, and the predictions are compared to real India VIX data. This may also be seen in code.

References:

Bantwa, A. (2017). A Study on India Volatility Index ( VIX ) and its Performance as Risk Management Tool in Indian Stock Market. Paripex-Indian Journal of Research, 6(1), 248–251.

Just, M., & Echaust, K. (2020). Stock market returns, volatility, correlation and liquidity during the COVID-19 crisis: Evidence from the Markov switching approach. Finance Research Letters, 37, 101775. https://doi.org/https://doi.org/10.1016/j.frl.2020.101775

Magner, N., Lavin, J. F., Valle, M., & Hardy, N. (2021). The predictive power of stock market’s expectations volatility: A financial synchronization phenomenon. PLOS ONE, 16(5), 1–21. https://doi.org/10.1371/journal.pone.0250846

Ozair, M. (2014). What Does the Vix Actually Measure? an Analysis of the Causation of Spx and Vix. ACRN Journal of Finance and Risk Perspectives, 3(2), 83–132.

2. One model can be good for fitting some data, but it is not meant to be suitable for all data. Therefore, before drawing certain conclusions, it is advisable to prove that the applied model is suitable through robustness or diagnostic tests. If the model used is not tested with certainty, it will lead to biased conclusions.

Response: We have gone through all of the preparatory tasks required to construct a robust model in this study. The first task is to see if the data is suitable for feeding into the model and, if it isn't, to try to make it. Now, both Econometric and machine learning models require stationary data, so we made the data stationary by taking the first differences of India VIX returns, and then we ran a statistical test to find potential structural breaks that could cause the data to become non-stationary. After that, we ran a test to see whether there were any ARCH effects, as GARCH-based models function best in presence of ARCH effects. After that, we needed to discover the best lag order in the data that would be utilized in the models, and we used AIC, BIC, and log-likelihood metrics to do so. After determining the lag order, we used GARCH models to fit the data.

We have already preprocessed the data while preparing it for GARCH models and we needed to find optimum hyperparameters for the LSTM model for which we used AIC & BIC metrics now, the most important thing to look for when making a robust machine learning model is the model's performance on training data & validation data, and we followed this procedure on different possible nodes, layers, and other hyperparameters possible while keeping in mind its performance on seen data (i.e. training data) & validation data.

Finally, the most common way for examining the robustness of trained models is to employ various forms of performance metrics, and we used three distinct types of metrics (RMSE, MAPE, and MDA) to verify the robustness of models on out of fold data. The three metrics used are fundamentally different from each other as RMSE tells the square root of the average standard deviation of predicted & actual values whereas MAPE tells the mean of deviation between predicted & actual values in terms of percentage, and MDA judges the models based on completely different parameter i.e. it compares the direction of predicted values & actual values. So, by combining all three metrics some robust conclusions can be drawn about the performance of models on the data which was provided.

Reviewer 3 Report

The revised version of the submission titled “Modeling and forecasting the volatility of NIFTY 50 using GARCH and RNN Models” (Manuscript ID: economies-1617927) improved in a constructive way. The author(s) responded to all the suggestions and recommendations expressed throughout the prior review round. Thus, the value of the paper heightened considerably. As such, I recommend paper acceptance in current form.

Author Response

We thank the reviewer for giving valuable suggestions which increased the quality of the research paper.